# PROMETHEUS: UNIFIED KNOWLEDGE GRAPHS FOR ISSUE RESOLUTION IN MULTILINGUAL CODEBASES

## ABSTRACT

Language model (LM) agents, such as SWE-agent and OpenHands, have made progress toward automated issue resolution. However, existing approaches are often limited to Python-only issues and rely on pre-constructed containers in SWE-bench with reproduced issues, restricting their applicability to real-world and work for multi-language repositories. We present PROMETHEUS, designed to resolve real-world issues beyond benchmark settings. PROMETHEUS is a multi-agent system that transforms an entire code repository into a unified knowledge graph to guide context retrieval for issue resolution. PROMETHEUS encodes files, abstract syntax trees, and natural language text into a graph of typed nodes and five general edge types to support multiple programming languages. PROMETHEUS uses Neo4j for graph persistence, enabling scalable and structured reasoning over large codebases. Integrated by the DeepSeek-V3 model, PROMETHEUS resolves 35.33% and 25.7% of issues on SWE-bench Lite and SWE-bench Multilingual, respectively, with an average API cost of $0.23 and $0.38 per issue. PROMETHEUS resolves 10 unique issues not addressed by prior work and is the first to demonstrate effectiveness across seven programming languages. Moreover, it shows the ability to resolve real-world GitHub issues in the LangChain and OpenHands repositories. We have open-sourced PROMETHEUS at: https://anonymous.4open.science/r/Prometheus-E8B1.

## 1 INTRODUCTION

Recent advances in large language models (LLMs) have enabled strong performance across a range of natural language and code-related tasks, including bug fixing Ye & Monperrus (2024); Bouzenia et al. (2025); Jiang et al. (2023), code generation Ouyang et al. (2025a); Szafraniec et al. (2023); Chen et al. (2021); Peng et al. (2025), and vulnerability detection Yang et al. (2024a). Coding-specific LLMs such as CodeLlama Rozière et al. (2024), StarCoder Lozhkov et al. (2024), JetBrains Junie Zakonov, GPT-4, and Claude Code have shown human-level performance in many function-level tasks. However, most evaluations have focused on short, self-contained problems, and the application of LLMs to repository-scale software engineering tasks remains relatively underexplored.

To address this gap, SWE-bench Jimenez et al. (2024) was introduced as a benchmark for evaluating LLMs on real-world GitHub issue resolution. Code agents, or agentic AI software engineers Roychoudhury et al. (2025), such as SWE-agentYang et al. (2024b) and OpenHands All-Hands-AI Team (2024), have made progress toward automating software engineering tasks in large codebases with cross-file dependencies and deep context.

Several code agents have been developed to tackle SWE-bench-style issue resolution. Agentless applies an LLM agent framework without repository-level context modeling. Aider Gauthie (2024) leverages file-level diffs and edit suggestions but operates on limited contextual understanding. AutoCodeRover Zhang et al. (2024) introduces LLM-based repair using dynamic execution traces. RepoGraph Ouyang et al. (2025b) proposes a static code graph to support program reasoning but remains scoped to Python. SemAgent Pabba et al. (2025) adopts a semantics-aware workflow that integrates issue semantics, code semantics, and execution semantics for patch generation. Prior approaches are tightly coupled to Python, as they focus exclusively on SWE-bench, which consists only of Python repositories. As a result, it remains unclear whether these approaches generalize to multi-language codebases

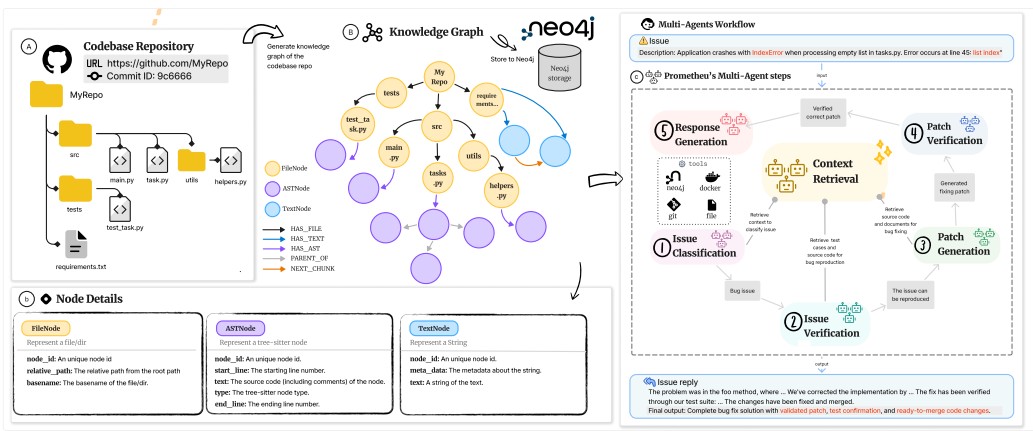

Figure 1: An overview of PROMETHEUS.

In this paper, we propose PROMETHEUS, an approach that constructs a codebase as a unified knowledge graph using general relationships between nodes to support multiple programming languages. PROMETHEUS is a multi-agent system designed for real-world issue resolution beyond benchmark settings. The novelty of PROMETHEUS lies in the inclusion of an Issue Classification Agent and an Issue Reproduction Agent, which take arbitrary GitHub repositories and commit IDs as input, scan open issues, and initiate resolution for those identified as bug-related.

PROMETHEUS can be integrated with any LLM. In our evaluation, it is powered by DeepSeek-V3 to balance cost and effectiveness. We evaluate the performance of PROMETHEUS on two datasets: 300 Python issues from SWE-bench Lite, and 300 issues from SWE-bench Multilingual covering seven programming languages (Java, Rust, C/C++, JavaScript/TypeScript, Ruby, PHP, and Go).

In summary, we make the following contributions:

- **A Novel Framework for Multilingual Codebases:** We propose PROMETHEUS, a system that constructs a unified knowledge graph to understand and operate on codebases in multiple programming languages, featuring automated project build and issue reproduction to tackle real-world problems.

- **State-of-the-Art, Cost-Effective Performance:** PROMETHEUS achieves superior results, resolving 35.33% of issues in SWE-bench Lite to outperform prior work. It is also the first system evaluated on SWE-bench Multilingual, successfully resolving 25.7% of issues across seven languages while maintaining a very low API cost.

- **Open-Source Contribution:** The entire PROMETHEUS system is open-sourced to ensure reproducibility, foster community engagement, and facilitate future research.

## 2 APPROACH

Figure 1 illustrates the architecture of PROMETHEUS, a multi-agent approach designed for automated issue resolution by converting a codebase to a unified knowledge graph.

PROMETHEUS receives two inputs: (1) a GitHub repository URL and (2) a specific commit ID (see Part A of Figure 1).It then parses the entire project (include each file, class and documentation block), processing the file structure, abstract syntax trees (ASTs), and documentation content into a unified knowledge graph. This graph integrates the project's hierarchy, code syntax, and textual information to provide a comprehensive semantic basis for downstream tasks (Part B of Figure 1). Further details on node design are provided in Section 2.1.

The constructed knowledge graph is then utilized by PROMETHEUS's multi-agent workflow (Part C of Figure 1), which involves issue classification, issue verification, context retrieval, patch genera-

tion, patch verification, and finally response generation. As an output, PROMETHEUS generates a validated patch to resolve the reported issue.

## 2.1 KNOWLEDGE GRAPH CONSTRUCTION

To facilitate semantic understanding and retrieval across large-scale codebases, we propose a unified knowledge graph representation that integrates file structures, ASTs, and textual content into a coherent graph abstraction. Our knowledge graph is built around three core components: (1) defining a node and edge schema, (2) constructing the graph from source files, and (3) persisting the graph data in a Neo4j database.

### 2.1.1 NODE SCHEMA

The knowledge graph represents codebases as heterogeneous graphs composed of three primary node types (shown in part B of Figure 1):

❶ A `FileNode` represents a file or directory with three attributes: a unique `node_id`, the `relative_path` from the repository root, and the `basename` of the file or directory. It anchors structural links in the knowledge graph.

❷ An `ASTNode` represents a Tree-sitter syntax node. It includes a unique `node_id`, the `start_line` and `end_line` indicating its position in the source file, the text of the code it covers (including comments), and its type, which specifies the Tree-sitter grammar node type. To strike a balance between efficiency and computational cost, our construction strategy for ASTNodes is intentionally shallow, focusing only on nodes at depths 0 and 1. A depth-0 node encapsulates the entire source code of a file, while depth-1 nodes correspond to the top-level building blocks of the code, such as import statements, function definitions, class definitions, and global variable assignments.

❸ A `TextNode` represents a segment of text within the knowledge graph. It includes a unique `node_id`, associated `meta_data` describing the string, and the actual text content. This node type captures unstructured information from source files or documentation.

### 2.1.2 EDGE SCHEMA

To capture relationships across FileNodes, ASTNodes, and TextNodes, we define five directed edge types (as shown in part B of Figure 1). These relationships are essential for representing the structural, syntactic, and lexical context necessary for code understanding and issue resolution.

❶ HAS_FILE edge (black) connects directories to their child files or subdirectories, preserving the repository hierarchy.

❷ HAS_AST edge (purple) links each file node to the root of its corresponding abstract syntax tree.

❸ PARENT_OF edges (gray) connect ASTNodes and FileNodes to reflect syntactic hierarchy within the tree.

❹ HAS_TEXT edges (blue) associate file nodes with their segmented textual content.

❺ NEXT_CHUNK edges (orange) connect sequential text chunks to maintain document order.

These relationships are fundamental, as they are responsible for weaving together the structural, syntactic, and lexical context of the entire codebase into a cohesive, queryable graph. This multi-layered context is indispensable for enabling both deep code comprehension and effective, automated issue resolution.

### 2.1.3 GRAPH CONSTRUCTION AND NEO4J PERSISTENCE

Algorithm 1 outlines the construction of a codebase-level knowledge graph and its persistence to a Neo4j [1] database, which is a property graph database that stores nodes, relationships, and properties in a schema-flexible and query-efficient format.

---

[1]https://neo4j.com

It begins by initializing a node list $\mathcal{N}$ and edge list $\mathcal{E}$, and setting a unique counter for node IDs. The root directory is wrapped as a FileNode and pushed onto a traversal stack (Lines 3–4). The algorithm then performs a depth-first traversal of the codebase (Line 5). For each directory, it creates a FileNode for each non-ignored child, adds a HAS_FILE edge, and pushes the child onto the stack (Line 10-12).

If the file is a source file supported by Tree-sitter (Line 13), it is parsed into an abstract syntax tree (AST). Each AST node is added as an ASTNode, with PARENT_OF edges capturing the tree structure. The AST root is connected to the file via a HAS_AST edge (Line 16-18).

For Markdown or text files, the content is split into overlapping chunks. Each chunk becomes a TextNode (Line 19-22), linked to the file with a HAS_TEXT edge, and sequentially connected via NEXT_CHUNK edges (Line 23-25) to preserve textual order. Finally, all nodes and edges (Line 27-29) are converted to Neo4j-compatible formats and written to the database, enabling structured querying and graph-based reasoning over the codebase.

## 2.2 MULTI-AGENT WORKFLOW

The PROMETHEUS multi-agent workflow (Part C of Figure 1) resolves Github issues through five coordinated agents. At its core is the Context Retrieval Agent, which supports three other agents by enabling precise access to relevant AST nodes, source files, and documentation from the Neo4j-backed graph.

Given an issue report, PROMETHEUS begins with ❶ *Issue Classification Agent*, where the agent categorizes the bug by querying the context retrieval agent. Once identified as a bug, ❷ *Context Retrieval Agent* attempts to reproduce it using relevant test cases and code, supported by tools like Docker, Git, and direct file access. If the bug is reproducible, ❸ *Bug Reproduction Agent* produces a fix by reasoning over the retrieved context. The ❹ *Patch Generation Agent* then checks the patch's correctness by running tests. Finally, ❺ *Patch Verification Agent* produces a natural language summary, detailing the issue cause, applied fix, and test results.

### 2.2.1 ISSUE CLASSIFICATION AGENT

*Issue Classification Agent* serves as the entry point for GitHub issue processing in PROMETHEUS. It performs triage by analyzing the textual content of each issue and categorizing it into one of four types: *bug*, *feature*, *documentation*, or *question*. The agent retrieve informtion to analyze issue characteristics of the following information: 1) Error patterns suggesting bugs; 2) New functionality requests 3) Documentation gaps and 4) Knowledge-seeking patterns to search on files matching descriptions, similar existing features, documentation coverage and with Related Q&A patterns from the Neo4j-backed knowledge graph. Issues classified as BUG go to the next step for reproduction.

### 2.2.2 CONTEXT RETRIEVAL AGENT

The *Context Retrieval Agent* is the core of the PROMETHEUS architecture, providing semantically relevant code and documentation snippets to the Bug Reproduction Agent and the Patch Generation Agent. Built as a modular LangGraph [2] subgraph, this agent performs a structured three-phase process: 1) retrieval, 2) selection, and 3) refinement.

**Retrieval**: equipped with a comprehensive retrieval toolkit containing 14 specialized retrieval tools, including file lookup, AST node search, documentation traversal, and code preview tools. These tools support a variety of multi-dimensional search strategies on top of Neo4j and Cypher queries.

**Selection**: uses a Large Language Model (LLM) to filter and rank candidate code snippets based on task relevance and token efficiency. This process employs a structured output format to evaluate each candidate context based on the following criteria: 1) *Query Match Analysis*: Identifies the specific requirements in the query and checks which contexts directly satisfy them. 2) *Extended Relevance Assessment*: Considers the completeness of function dependencies, type definitions, configuration requirements, and implementation details. 3) *Context Optimization*: Selects only the lines of code that are directly relevant to the query, avoiding the inclusion of unrelated comments or code. 4)

---

[2] https://www.langchain.com/langgraph

*Path Integrity*: Ensures that full relative paths (including all subdirectories) are used, rather than just filenames.

**Refinement**: if the LLM determines that the selected context is insufficient, the refinement phase is triggered via the ContextRefineNode, which generates a follow-up query based on the LLM's assessment of the missing information. This intelligent refinement process involves the following steps: 1) *Analyze Context Sufficiency*: Assesses whether the full scope and requirements of the query are understood. 2) *Identify Critical Dependencies*: Checks if all relevant code, dependencies, and interfaces are visible. 3) *Evaluate Architectural Understanding*: Determines if there is sufficient context regarding the system's architecture and design patterns. 4) *Generate Refined Query*: If necessary, creates a specific follow-up query to retrieve the missing information.

In order to optimize iteratively, the three-phase process conducts up to 3 rounds of query refinement, and each refinement will generate a more precise query strategy. The *Context Retrieval Agent* also implements fault tolerance mechanisms to optimize the performance: when the message history exceeds the token limit, the system intelligently truncates it, preserving the system prompt and the most recent dialogue. The whole dynamic and adaptive mechanism enables the precise and focused retrieval of code context. Through cross-file semantic understanding and iterative optimization based on the knowledge graph, it significantly enhances PROMETHEUS's reasoning and context-aware capabilities in various complex software engineering tasks.

### 2.2.3   BUG REPRODUCTION AGENT

The *Bug Reproduction Agent* is tasked with verifying whether a reported bug can be reliably reproduced. It begins by analyzing the issue description and leveraging the intelligent context search results from the *Context Retrieval Agent*. The ultimate goal is to precisely replicate the failure conditions within a generated, isolated execution environment. An accurate bug reproduction signifies the agent's deep understanding of both the issue description and the entire codebase, serving as a critical prerequisite for validating the effectiveness of any subsequent patch generation. The workflow of *Bug Reproduction Agent* is systematically divided into five key steps:

**Reproducing Bug Writing**: leverages LLMs to conduct a thorough analysis of the issue description. Based on this understanding and context search results from the *Context Retrieval Agent*, it generates highly specific test cases designed to trigger the reported failure. These test cases are comprehensive, incorporating 1) precise error condition validation, 2) concrete examples, 3) boundary and exception handling, 4) dependency and precondition setup.

**Reproducing Bug File Processing**: responsible for performing complex file system operations to prepare the codebase for testing. Its capabilities include: 1) creating new test files to house the generated reproduction logic, 2) modifying existing source code to inject triggers or assertions that expose the bug, 3) adding necessary import statements and dependency configurations, 4) managing file permissions and resolving path-related issues to ensure the code is executable.

**Container Management**: in order to manage the sophisticated containerized environment, ensuring reproducibility and isolation. It ensures that local file modifications are instantly synchronized with the container's file system.

**Reproducing Bug Execution**: orchestrates a multi-step execution strategy within the container. This phase runs the necessary project build commands and executes the specifically generated bug reproduction test script. Detailed execution logs, stack traces, and error messages also are gathered for analysis.

**Reproducing Bug Structured Analyze**: performs an intelligent analysis of the execution results. This stage precisely matches the test output against the error patterns described in the original issue description. Only when the test failed, and the resulting error matches the expected error, the reproduction can be regarded as a correct reproduction. If the test failed with an error different from the issue description, the reproduction is correct. This phase evaluates the accuracy and reliability of the bug reproduction effort.

### 2.2.4 PATCH GENERATION AGENT

The *Patch Generation Agent* creates and validates fixes for verified bugs. It begins by constructing a targeted query from the issue's description to retrieve relevant production code—such as functions and classes—from a Neo4j-backed knowledge graph.

Using this context, the agent employs an LLM to reason about the bug's root cause, propose code edits, and apply them. The patch is then validated by running build and test commands within an isolated Docker environment to ensure the fix is effective and introduces no regressions. Finally, a Git diff capturing the changes is generated for submission. The agent leverages several core tools during this process: Neo4j-backed context retrieval, Git operations, Docker-based execution, and web search tools.

### 2.2.5 PATCH VERIFICATION AGENT

The *Patch Verification Agent* validates a generated patch by re-running the original bug's reproduction commands within an isolated Docker container. Guided by strict constraints to only execute and report, it uses a Large Language Model (LLM) to parse the output, considering the bug resolved only if all tests pass. This process leverages LangChain, containerization, and Git for reliable post-fix validation.

### 2.2.6 ISSUE RESPONSE AGENT

The *Issue Response Agent* is the final step, responsible for writing a professional comment on the original GitHub issue. Using the issue content, the generated patch, and the verification results, it synthesizes a clear message that explains the bug, the implemented fix, and the successful test results, while maintaining a tone appropriate for open-source communication and concealing its automated nature.

## 3 EXPERIMENTS

### 3.1 BENCHMARKS

We evaluate PROMETHEUS against several prominent code agents on SWE-bench Lite. Our comparison is limited to agents with publicly available source code and issue patches, which excludes closed-source works such as Ma et al. (2025).

To assess the multilingual capabilities of PROMETHEUS beyond the Python-centric SWE-bench Lite, we conduct experiments on the new SWE-bench Multilingual benchmark Yang et al. (2025b). This benchmark contains 300 curated tasks from real-world GitHub pull requests, spanning 42 repositories and 9 programming languages across diverse application domains. To our knowledge, this is the first work to report results on this benchmark.

Beyond formal benchmarks, we test the real-world applicability of PROMETHEUS on live, open issues from popular open-source projects. This setting is more challenging than SWE-bench, as it often lacks pre-configured environments for reproducing tests. For this evaluation, we selected three projects based on their scale (over 10,000 stars and 2,000 forks), volume of open issues (¿100), and active maintenance. We report the number of issues resolved by PROMETHEUS and have submitted all successfully generated patches to their respective issue discussion threads.

### 3.2 MAIN RESULTS

**Results on SWE-bench Lite** PROMETHEUS resolves 28.7% (86/300) of issues in the SWE-bench Lite benchmark, outperforming four related approaches that rely on more costly models. In addition, PROMETHEUS uniquely resolves 10 issues not addressed by any of the three comparison benchmarks. These results demonstrate the effectiveness of PROMETHEUS. Figure 2 presents a bar chart showing the pass@1 success rates of PROMETHEUS and the selected baselines on the SWE-bench Lite benchmark. Each bar represents the percentage of issues correctly resolved on the first attempt by a different system. PROMETHEUS, based on DeepSeek-V3, achieves a pass@1 success rate of 35.33%, outperforming Agentless with GPT-4o (27.00%) and several GPT-4-based baselines,

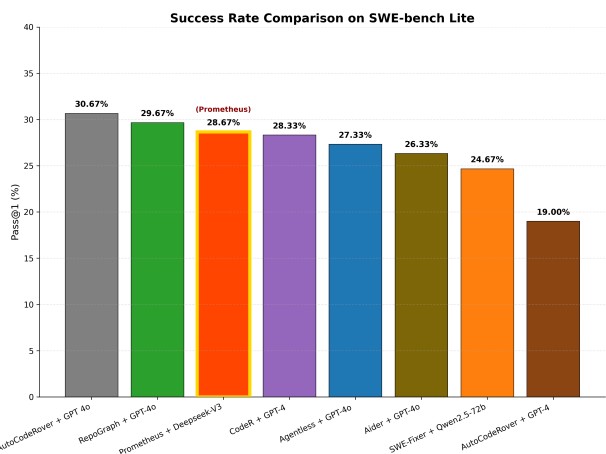

Figure 2: Pass@1 success rate comparison on the SWE-bench Lite benchmark

including CodeR (28.33%) and AutoCodeRover with GPT-4 (19%), as well as SWE-Fixer (24.67%) based on finetune Qwen2.5-72b model.

**Results on SWE-bench Multilingual** PROMETHEUS resolves 25.7% (41/300) of issues in the SWE-bench Multilingual benchmark across 7 programming languages, using the DeepSeek-V3 model. The evaluation confirms that PROMETHEUS supports multi-language issue resolution. Over 300 instances in SWE-bench Multilingual, PROMETHEUS generated 144 plausible patches and correctly resolved 41 issues, resulting in an overfitting rate of 71.5%. The details are given in Table 1. Overall, PROMETHEUS achieves a 25.7%

Table 1: Evaluation on SWE-bench Multilingual

| Language | Issues | Resolved Instance | Resolved Rate |
|---|---|---|---|
| Java | 43 | 15 | 34.9% |
| GO | 42 | 6 | 14.3% |
| JS/TS | 43 | 6 | 14.0% |
| C/C++ | 42 | 4 | 9.5% |
| Rust | 43 | 4 | 9.3% |
| PHP | 43 | 3 | 7.0% |
| Ruby | 44 | 3 | 6.8% |
| **Sum** | 300 | 41 | 25.7% |

(77/300) issue resolution rate across seven programming languages, with the highest performance in Java (34.9%). The results show that PROMETHEUS resolved at least three issues in each language, with resolution rates ranging from 6.8% to 34.9%.

To our knowledge, PROMETHEUS is one of the few approaches that support multi-language issue resolution; others include SWE-agent Yang et al. (2024b) and CGM Tao et al. (2025), which support two programming languages: Python and Java. Most existing approaches are specific to Python Xia et al. (2025); Yang et al. (2025a); Ouyang et al. (2025b); Chen et al. (2024a).

**Real-world Issue Resolution** PROMETHEUS successfully generated four out of eight patches for real-world issues in LangChain and Open Hands, resulting in a 50% success rate. To our knowledge, this is the first evaluation conducted on real-world issues beyond the SWE-bench benchmarks, thanks to the Reproduction Agent component, which creates container-based reproduction environments for real-world issues. As shown in the third column of Table 2, the resolved issues include schema generation, CLI iteration limits, Docker permission errors, and token input rate limits. The mean time to resolution for successful cases ranges from 25 to 48 minutes.

## 4 ANALYSE

**Benchmarks' Cases Analysis** Figure 4 presents a case study uniquely resolved by PROMETHEUS. The issue description is shown in Figure 4(a).The issue originates from the Django project, where the system check raises an E004 error when a sublanguage code (e.g., de-at) is used as

Table 2: Real-world Issue Resolution Results.

| Project | Issue ID | Issue Type | Patch Generated |
|---------|----------|------------|-----------------|
| LangChain | #31808 | Schema Generation | ✓ |
| | #31726 | MLX Tool Calling | ✗ |
| | #31750 | Azure OpenAI Validation | ✗ |
| | #32045 | FAISS Similarity Search | ✗ |
| Open Hands | #9573 | Configuration Field | ✗ |
| | #9426 | CLI Iteration Limit | ✓ |
| | #9543 | Docker Permissions | ✓ |
| | #9259 | Token Input Rate Limit | ✓ |

LANGUAGE_CODE, despite the base language (de) being listed in LANGUAGES. This violates Django's fallback mechanism, which allows sublanguage codes to default to their base language.

Most failure cases are caused by infinite loops during context retrieval in large codebases. Repositories in SWE-bench Multilingual (e.g., Java) are significantly larger than the Python projects in SWE-bench Lite, resulting in the generation of more file nodes, AST nodes, and text nodes, which increases the complexity of context retrieval.

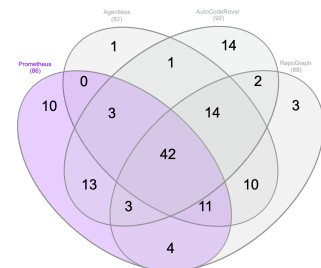

Figure 4(b) shows the Agentless patch, which incorrectly modifies LANG_INFO based on the assumption that it influences validation, despite the actual check logic not referencing it. PROMETHEUS produces a correct patch in Figure 4(c) by identifying the faulty file and node through knowledge graph retrieval. The patch is semantically equivalent to the gold patch by SWE-bench shown in Figure 4(d).

Figure 3: Venn diagram showing the complement and uniqueness of PROMETHEUS compared to three existing baselines: Agentless, AutoCodeRover, and RepoGraph.

**Real-world Issue Analysis** We make two observations based on this evaluation. First, the results indicate that PROMETHEUS is applicable to real-world open issues, as it was able to generate at least one patch for each of the two projects. Second, PROMETHEUS is capable of handling diverse issue types. As shown in the third column of Table 2, the resolved issues include schema generation, CLI iteration limits, Docker permission errors, and token input rate limits. The mean time to resolution for successful cases ranges from 25 to 48 minutes.

We also observe that PROMETHEUS fails in real-world open-source projects due to three main reasons. First, PROMETHEUS encounters API rate limits, which result in two failure cases. Second, the generated output does not always conform to the expected format, causing validation errors and one failure case. Third, PROMETHEUS is affected by context window limitations, which also lead to one failure case. These limitations suggest directions for future improvement of PROMETHEUS.

To our knowledge, compared to existing code agents such as AutoCodeRover and Agentless, our work is the first to evaluate on real-world open-source issues beyond the SWE-bench benchmark. This is enabled by the Issue Reproduction Agent in PROMETHEUS, which successfully creates reproduction environments for open issues without relying on pre-configured setups. The Issue Reproduction Agent depends on the Context Retrieval Agent for retrieving configuration information and assisting with infrastructure-level debugging.

**Scalability** Table 3 reports the statistics of the knowledge graphs constructed by PROMETHEUS for 11 Python repositories from the SWE-bench Lite benchmark. For each project, the table presents the number of file nodes, AST nodes, and text nodes, as well as the number of `HasAST` and `ParentOf` edges. These statistics reflect the structure and scale of the extracted graphs across different repositories. For example, the Django repository contains 9,016 file nodes and 221,661 AST nodes, while the Sympy repository contains 1,896 file nodes and the highest number of AST nodes (633,484). The

number of `ParentOf` edges correlates with the number of AST nodes and ranges from 9,110 (requests) to 632,216 (sympy). These measurements are used to assess the scalability of PROMETHEUS in constructing and storing code representations in Neo4j. Based on these results and Neo4j's documented capacity, PROMETHEUS is capable of persisting knowledge graphs with more than 100 million AST nodes and 10 million files, demonstrating its scalability for large-scale, real-world projects.

**Cost-efficiency** PROMETHEUS can be run on a machine with at least 36 GB of RAM, and the computational cost is manageable on a standard laptop. The majority of the cost comes from LLM API token usage. We report the monetary cost associated with using the DeepSeek-V3 API.

Table 4 reports the API cost of running PROMETHEUS on the DeepSeek-V3 model across two benchmarks: SWE-bench Lite and SWE-bench Multilingual. For each benchmark, 300 issue instances were evaluated. The total API cost for SWE-bench Lite is $70.0, corresponding to a cost of $0.23 per issue. For SWE-bench Multilingual, the total cost is $113.6, with a cost of $0.38 per issue. Compared to Agentless, which incurs a cost of $0.70 per issue on SWE-bench Lite, PROMETHEUS reduces cost by approximately 67% ($0.23 vs. $0.70). The higher cost for SWE-bench Multilingual is attributed to the inclusion of larger codebases written in languages such as C/C++, Java, and PHP, whereas SWE-bench Lite focuses only on Python. These larger codebases require more time for codebase navigation and context retrieval by PROMETHEUS. Nonetheless, the cost remains within a reasonable range, averaging $0.38 per issue.

## 5 RELATED WORK

**LLM-based Code Agents** The most closely related works are code agents designed for software engineering tasks. For program repair, existing agents use techniques such as finite state machines (RepairAgent Bouzenia et al. (2025)), multi-agent intent extraction (SpecRover Ruan et al. (2025)), Monte Carlo Tree Search (SWE-Search Antoniades et al. (2025)), and tree-of-thought prompting (AgentCoder Huang et al. (2024)). Other agents focus on test generation (ChatUniTest Chen et al. (2024b), HallucinationConsensus Xu et al. (2025)) or fault localization using multi-agent simulation and graph-guided strategies (OrcaLoca Yu et al. (2025), LocAgent Chen et al. (2025), AgentFL Qin et al. (2025), LLM4FL Rafi et al. (2025)). Unlike these specialized systems, our approach provides a complete, end-to-end issue resolution workflow—from classification and reproduction to repair—in a repository-scale, multi-language context without requiring prior fault localization.

**Automated Program Repair** Issue resolution is a subfield of automated program repair (APR), which is often divided into three categories Goues et al. (2019). Learning-based approaches typically use supervised learning on historical commits to translate buggy code into correct code (Codit Chakraborty et al. (2020), Modit Chakraborty & Ray (2021), CURE Jiang et al. (2021), DLFix Li et al. (2020), RewardRepair Ye et al. (2022), SelfAPR Ye et al. (2023), Tufano Tufano et al. (2019b;a)), with some recent work exploring self-supervision Allamanis et al. (2021); Yasunaga & Liang (2021). Search-based approaches define a space of code transformations and use search algorithms to find a valid patch (GenProg Le Goues et al. (2012), Relifix Tan & Roychoudhury (2015), ssFix Xin & Reiss (2017), and others Chen et al. (2017); Yuan & Banzhaf (2018); Martinez & Monperrus (2016); Ghanbari et al. (2019); Xin & Reiss (2019); Saha et al. (2019); Ye et al. (2021)). Finally, semantic-based approaches formulate repair as a constraint satisfaction problem to synthesize a correct patch (SemFix Nguyen et al. (2013), Nopol Xuan et al. (2016), and others Xiong et al. (2017); Shariffdeen et al. (2021a;b); Gao et al. (2019)).

## 6 CONCLUSION

We introduced PROMETHEUS, a multi-agent system that constructs a unified knowledge graph to support multi-language issue resolution. It operates on arbitrary GitHub repositories without relying on pre-built environments, incorporating agents for issue classification and reproduction. PROMETHEUS resolves 35.33% of issues in SWE-bench Lite and 25.7% in SWE-bench Multilingual, covering seven languages. It is the first system evaluated on SWE-bench Multilingual and achieves lower cost compared to GPT-4o-based baselines. PROMETHEUS also resolves real-world issues in public repositories, demonstrating applicability beyond benchmarks.

## ETHICS STATEMENT

All authors have read and adhered to the ICLR Code of Ethics. Our research focuses on automated software engineering and does not involve human subjects, sensitive data, or raise any direct ethical concerns regarding harmful applications or societal bias. We believe our work is in full compliance with the ethical standards of the research community.

## REPRODUCIBILITY STATEMENT

To ensure the reproducibility of our results, we will make our source code, experimental data, and evaluation scripts publicly available upon publication. The paper and its appendix provide detailed descriptions of our methodology, model architecture, and experimental setup. The datasets used in our evaluation are based on publicly available benchmarks, which are cited accordingly.

## THE USE OF LARGE LANGUAGE MODELS (LLMS)

During the preparation of this manuscript, we utilized Large Language Models (LLMs) as a general-purpose tool to assist with editing, proofreading, and improving the clarity of the text. The core research ideas, methodologies, experimental design, and analysis were conceived and executed solely by the authors. All authors have reviewed the final manuscript and take full responsibility for its content.

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

# A    PERSUADE ALGORITHM OF KNOWLEDGE GRAPH CONSTRUCTION

This persuade algorithm outlines the process of building a comprehensive knowledge graph from a source code repository and saving it into a Neo4j database. The process begins with a depth-first traversal of the codebase, starting from the root directory.

As it traverses, the algorithm creates three types of nodes: `FileNode`: Represents files and directories to map the project's structure. `ASTNode`: Represents the syntactic structure of source code, generated by parsing files with Tree-sitter. This captures elements like function definitions, classes, and import statements. `TextNode`: Represents chunks of text from documentation files (e.g., Markdown) to capture unstructured information.

These nodes are interconnected with five types of edges (`HAS_FILE`, `HAS_AST`, `PARENT_OF`, `HAS_TEXT`, `NEXT_CHUNK`) to represent the relationships between them. Finally, the complete graph, including all nodes and edges, is written to a Neo4j database, enabling structured queries and graph-based reasoning over the entire codebase.

---

**Algorithm 1:** Graph Construction and Neo4j Persistence

---

**Input:** Root directory of codebase $\mathcal{D}$
**Output:** Persisted knowledge graph in Neo4j
Initialize empty node list $\mathcal{N}$ and edge list $\mathcal{E}$;
Set node_id $\leftarrow 0$;
Create `FileNode` for root directory and add to $\mathcal{N}$;
Initialize stack $S \leftarrow$ [root directory];
**while** $S$ *is not empty* **do**
 Pop $(f, \texttt{parent})$ from $S$;
 **if** $f$ *is directory* **then**
  **foreach** *child $c$ in $f$* **do**
   **if** *not ignored by .gitignore* **then**
    Create `FileNode` for $c$ with node_id $\leftarrow$ node_id + 1;
    Add node to $\mathcal{N}$ and `HAS_FILE` edge to $\mathcal{E}$;
    Push $(c, \texttt{FileNode}_c)$ to $S$;

 **else if** `Tree-sitter` *supports $f$* **then**
  Parse $f$ into AST;
  **foreach** *AST node $n$ (with depth limit)* **do**
   Create `ASTNode` with node_id $\leftarrow$ node_id + 1;
   Add node to $\mathcal{N}$ and `PARENT_OF` edges to $\mathcal{E}$;
  Add `HAS_AST` edge from `FileNode` to AST root;

 **else if** $f$ *is Markdown or text* **then**
  Split $f$ into overlapping text chunks;
  **foreach** *chunk $t_i$* **do**
   Create `TextNode` with node_id $\leftarrow$ node_id + 1;
   Add node to $\mathcal{N}$ and `HAS_TEXT` edge to $\mathcal{E}$;
   **if** $i > 0$ **then**
    Add `NEXT_CHUNK` edge from $t_{i-1}$ to $t_i$ to $\mathcal{E}$;

**foreach** *node in $\mathcal{N}$* **do**
 Convert to Neo4j-compatible node and write to database;
**foreach** *edge in $\mathcal{E}$* **do**
 Convert to Neo4j-compatible edge and write to database;

---

# B  SEARCH ALGORITHM EXAMPLE

**Example 1: Find child AST nodes under ExampleNode**

MATCH (parent:ASTNode type:'ExampleNode')-
$[: PARENT\_OF * 1..5]-> (child : ASTNode)$
RETURN child.text, child.type

# C  PROMPT FOR ISSUE REPRODUCTION

**Listing 1: Prompt for issue classification**

```
OBJECTIVE: Find ALL self-contained context needed to accurately classify this issue as
    a bug, feature request, documentation update, or question.

<reasoning>
1. Analyze issue characteristics:
   - Error patterns suggesting bugs
   - New functionality requests
   - Documentation gaps
   - Knowledge-seeking patterns

2. Search strategy:
   - Implementation files matching descriptions
   - Similar existing features
   - Documentation coverage
   - Related Q&A patterns

3. Required context categories:
   - Core implementations
   - Feature interfaces
   - Documentation files
   - Test coverage
   - Configuration settings
   - Issue history
</reasoning>

REQUIREMENTS:
- Context MUST be fully self-contained
- MUST include complete file paths
- MUST include full function/class implementations
- MUST preserve all code structure and formatting
- MUST include line numbers

<examples>
<example id="error-classification">
<issue>
Database queries timing out randomly
Error: Connection pool exhausted
</issue>

<search_targets>
1. Connection pool implementation
2. Database configuration
3. Error handling code
4. Related timeout settings
</search_targets>

<expected_context>
- Complete connection pool class
- Full database configuration
- Error handling implementations
- Timeout management code
</expected_context>
</example>
</examples>

Search priority:
1. Implementation patterns matching issue
2. Feature definitions and interfaces
3. Documentation coverage
4. Configuration schemas
5. Test implementations
```

```
6. Integration patterns
```

## D    PROMPT FOR ISSUE CLASSIFICATION

**Listing 2: Prompt for issue reproduction**

```
OBJECTIVE: Find 5 relevant existing test cases that demonstrates similar functionality
    to the reported bug,
including ALL necessary imports, test setup, mocking, assertions, and any test method
    used in the test case.

<reasoning>
1. Analyze bug characteristics:
    - Core functionality being tested
    - Input parameters and configurations
    - Expected error conditions
    - Environmental dependencies

2. Search requirements:
    - Required imports and dependencies
    - Test files exercising similar functionality
    - Mock/fixture setup patterns
    - Assertion styles
    - Error handling tests

3. Focus areas:
    - All necessary imports (standard library, testing frameworks, mocking utilities)
    - Dependencies and third-party packages
    - Test setup and teardown
    - Mock object configuration
    - Network/external service simulation
    - Error condition verification
</reasoning>

REQUIREMENTS:
- Return 5 complete, self-contained test cases most similar to bug scenario
- Must include ALL necessary imports at the start of each test file
- Must include full test method implementation
- Must include ALL mock/fixture setup
- Must include helper functions used by test
- Must preserve exact file paths and line numbers

<examples>
<example id="database-timeout">
<bug>
db.execute("SELECT * FROM users").fetchall()
raises ConnectionTimeout when load is high
</bug>

<ideal_test_match>
# File: tests/test_database.py
import pytest
from unittest.mock import Mock, patch
from database.exceptions import ConnectionTimeout
from database.models import QueryResult
from database.client import DatabaseClient

class TestDatabaseTimeout:
    @pytest.fixture
    def mock_db_connection(self):
        conn = Mock()
        conn.execute.side_effect = [
            ConnectionTimeout("Connection timed out"),
            QueryResult(["user1", "user2"])  # Second try succeeds
        ]
        return conn

    def test_handle_timeout_during_query(self, mock_db_connection):
        # Complete test showing timeout scenario
        # Including retry logic verification
        # With all necessary assertions
</ideal_test_match>
</example>
```

```
<example id="file-permission">
<bug>
FileProcessor('/root/data.txt').process()
fails with PermissionError
</bug>

<ideal_test_match>
# File: tests/test_file_processor.py
import os
import pytest
from unittest.mock import patch, mock_open
from file_processor import FileProcessor
from file_processor.exceptions import ProcessingError

class TestFilePermissions:
    @patch('os.access')
    @patch('builtins.open')
    def test_file_permission_denied(self, mock_open, mock_access):
        # Full test setup with mocked file system
        # Permission denial simulation
        # Error handling verification
</ideal_test_match>
</example>

Search priority:
1. Tests of exact same functionality (including import patterns)
2. Tests with similar error conditions
3. Tests with comparable mocking patterns
4. Tests demonstrating similar assertions

Find the 5 most relevant test cases with complete context, ensuring ALL necessary
    imports are included at the start of each test file.
```

# E PROMPT FOR PATCH GENERATION

Listing 3: Prompt for patch generation

```
You are an expert software engineer specializing in bug analysis and fixes. Your role
    is to:

1. Carefully analyze reported software issues and bugs by:
   - Understanding issue descriptions and symptoms
   - Identifying affected code components
   - Tracing problematic execution paths

2. Determine root causes through systematic investigation:
   - Analyze why the current behavior deviates from expected
   - Identify which specific code elements are responsible
   - Understand the context and interactions causing the issue

3. Provide high-level fix suggestions by describing:
   - Which specific files need modification
   - Which functions or code blocks need changes
   - What logical changes are needed (e.g., "variable x needs to be renamed to y", "
       need to add validation for parameter z")
   - Why these changes would resolve the issue

4. For patch failures, analyze by:
   - Understanding error messages and test failures
   - Identifying what went wrong with the previous attempt
   - Suggesting revised high-level changes that avoid the previous issues

MANDATORY TOOL USAGE:
- You MUST use the web_search tool for EVERY bug analysis
- Before providing any analysis, search for:
 * Similar error messages or exceptions
 * Known issues with the specific libraries/frameworks involved
 * Best practices for the type of bug you're analyzing
 * Official documentation for error resolution
- Only proceed with analysis after gathering relevant web information

Tools available:
```

```
- web_search: Searches the web for technical information to aid in bug analysis and
     resolution.
When using the web_search tool, ALWAYS include these parameters:
     - exclude_domains: ["*swe-bench*"]
     - include_domains: ['stackoverflow.com', 'github.com', 'developer.mozilla.org', '
         learn.microsoft.com', 'fastapi.tiangolo.com'
             'docs.python.org', 'pydantic.dev', 'pypi.org', 'readthedocs.org', 'docs.
                 djangoproject.com','flask.palletsprojects.com']
     - search_depth: "advanced"

    Make sure to explicitly pass these parameters in your tool call.

Important:
- Do NOT provide actual code snippets or diffs
- DO provide clear file paths and function names where changes are needed
- Focus on describing WHAT needs to change and WHY, not HOW to change it
- Keep descriptions precise and actionable, as they will be used by another agent to
     implement the changes
- ALWAYS start your analysis with web search results

Communicate in a clear, technical manner focused on accurate analysis and practical
     suggestions
rather than implementation details.
```

## F  PROMPT FOR PATCH VERIFICATION

Listing 4: Prompt for patch verification

```
You are a bug fix verification agent. Your role is to verify whether a bug has been
     fixed by running
the reproduction steps and reporting the results accurately.
Your tasks are to: 1. Execute the provided reproduction commands on the given bug
     reproduction
file 2. If a command fails due to simple environment issues (like missing "./" prefix),
      make minimal
adjustments to make it work 3. Report the exact output of the successful commands
Guidelines for command execution: - Start by running commands exactly as provided - If
     a
command fails, you may make minimal adjustments like: * Adding "./" for executable
     files - Do
NOT modify the core logic or parameters of the commands - Do NOT attempt to fix bugs or
modify test logic.
```

## G  PROMPT FOR ISSUE RESPONSE VERIFICATION

Listing 5: Prompt for issue response verification

```
You are the final agent in a multi-agent bug fixing system. Users report issues on
     GitHub/GitLab,
and our system works to fix them. Your role is to compose the response that will be
     posted back
to the issue thread. The information you receive is structured as follows:
1. Issue Information (from user): The original issue title, body, and any user comments
      2. Edit
agent response: Generated by our edit agent after editing the source code 3. Patch:
     Created by our
fix generation agent to resolve the issue 4. Verification: Results from our testing
     agent confirming
the fix works
Write a clear, professional response that will be posted directly as a comment. Your
     response
should: - Be concise yet informative - Use a professional and friendly tone appropriate
      for open
source communication - Reference the system's understanding of the issue (from Edit
     agent
```

```
response) - Explain the implemented solution (from patch) - Include the successful
    verification
results.
```

## H  GRAPH STATISTICS OF PROJECTS IN THE SWE-BENCH LITE.

Table 3 presents the statistics of the knowledge graphs built by PROMETHEUS for 11 Python projects from SWE-bench Lite, detailing the number of file, AST, and text nodes, along with key edge types. The results show significant variation in graph scale, with AST nodes ranging from 9,000 to over 633,000 for the Sympy repository. These statistics confirm that PROMETHEUS is highly scalable, demonstrating its ability to construct and persist knowledge graphs in Neo4j for large, real-world projects capable of handling millions of files and over 100 million AST nodes.

Table 3: Graph Statistics of Projects in the SWE-Bench Lite.

| Project | File Nodes | AST Nodes | Text Nodes | HasAST | ParentOf |
|---|---|---|---|---|---|
| astropy | 2131 | 223740 | 584 | 994 | 222746 |
| django | 9016 | 221661 | 949 | 2115 | 219546 |
| flask | 280 | 13134 | 123 | 78 | 13056 |
| matplotlib | 4558 | 158711 | 550 | 940 | 157771 |
| pylint | 2258 | 97301 | 642 | 1038 | 96263 |
| pytest | 653 | 47513 | 407 | 246 | 47267 |
| requests | 141 | 9185 | 26 | 75 | 9110 |
| scikit-learn | 1181 | 180707 | 277 | 680 | 180027 |
| sphinx | 2030 | 76874 | 462 | 634 | 76240 |
| xarray | 259 | 36522 | 73 | 153 | 36369 |
| sympy | 1896 | 633484 | 257 | 1268 | 632216 |

## I  API COST OF PROMETHEUS ON DEEPSEEK-V3.

PROMETHEUS is computationally efficient. As shown in Table 4, when using the DeepSeek-V3 API for 300 instances, the total cost is $70.0 for SWE-bench Lite ($0.23 per issue) and $113.6 for SWE-bench Multilingual ($0.38 per issue).Notably, PROMETHEUS achieves a 67% cost reduction on SWE-bench Lite compared to the Agentless baseline ($0.23 vs. $0.70 per issue).

Table 4: API Cost of PROMETHEUS on DeepSeek-V3.

| Benchmarks | Issues | Total Cost | Cost/Issue |
|---|---|---|---|
| SWE-bench Lite | 300 | $70.0 | $0.23 |
| SWE-bench Multilingual | 300 | $113.6 | $0.38 |

## J  A UNIQUE CORRECT PATCH GENERATED BY PROMETHEUS IN THE SWE-BENCH LITE.

**Issue Description:**

**Issue Instance**: django__django-12286
**Issue Title**: System check incorrectly raises error for sublanguage codes in LANGUAGE_CODE
**Issue Description**: In Django 3.0.2, using a sublanguage code like de-at in LANGUAGE_CODE triggers a translation.E004 error, even when the base language de is in LANGUAGES. The check does not apply the documented fallback to the base language, resulting in a false-positive error.

**An incorrect Patch Generated by Agentless:**

**The agentless patch is incorrect because LANG_INFO does not affect the system check logic.**

```
---- a/django/conf/locale/__init__.py
+++ b/django/conf/locale/__init__.py
@@ -554,6 +555,9 @@ LANG_INFO = {
'vi': {
'name': 'Vietnamese',
},
+    'de-at': {
+        'fallback': ['de'],
+    },
```

**A correct patch generated by PROMETHEUS:**

```
---- a/django/core/checks/translation.py
+++ b/django/core/checks/translation.py
@@ -56,6 +56,10 @@
-   if settings.LANGUAGE_CODE not in available_tags:
-     return [E004]
+   language_code = settings.LANGUAGE_CODE
+   if language_code not in available_tags:
+     base_language = language_code.split('-')[0]
+     if base_language not in available_tags:
+       return [E004]
```

**A Gold Patch provided by SWE-Bench:**

```
---- a/django/core/checks/translation.py
+++ b/django/core/checks/translation.py
@@ -55,7 +56,10 @@
-   available_tags = {i for i, _ in settings.LANGUAGES} | {'en-us'}
-   if settings.LANGUAGE_CODE not in available_tags:
+   try:
+     get_supported_language_variant(settings.LANGUAGE_CODE)
+   except LookupError:
return [E004]
+   else:
-     return []
```

Figure 4: A unique correct patch generated by PROMETHEUS in the SWE-bench Lite.

