# OpenReview forum: "Prometheus: Unified Knowledge Graphs for Issue Resolution in Multilingual Codebases"
_ICLR.cc/2026/Conference — Submitted to ICLR 2026_

### Official Review · Reviewer_2dHA · 2025-10-28

**Soundness:** 1
**Presentation:** 3
**Contribution:** 1
**Rating:** 2
**Confidence:** 5

**Summary:**

The paper proposes an agent scaffold for a software engineering agent with the particular focus on a novel retrieval component that constructs a graph knowledge base connecting files and AST nodes. The system leverages DeepSeek-V1 and achieves comparable performance to other agent scaffolds using GPT-4o.

**Strengths:**

- The paper clearly describes the proposed agent implementation.
- The proposed system achieves good performance on both SWE-bench and SWE-bench-multilingual.

**Weaknesses:**

- The individual components of the proposed system are not studied separately, e.g., it is unclear how much other scaffolds would perform with DeepSeek-V3 or how the Prometheus scaffold would change the performance with other base models.
- The model primarily compares to models that perform quite poorly on SWE-bench, as opposed to for example GPT-5, Sonnet-4, Sonnet-4.5, Qwen-Coder. This makes it hard to understand whether the context retrieval component is relevant with state-of-the-art approaches.
- The claim of “novel” solved issues is concerning since (a) it does not seem to consider more recent approaches and (b) the pass/fail of issues is inherently stochastic and could be due to random sampling (in the absence of pass@k analysis).

**Questions:**

- How would the system perform with Sonnet-4.5 or Qwen Coder as a base model?
- How does the system perform compared when ablating the proposed context retrieval component with context sub-agents (e.g. popular in Claude Code) or RAG systems (e.g. SWE-search)?

---

### Official Review · Reviewer_4Reo · 2025-10-31

**Soundness:** 2
**Presentation:** 3
**Contribution:** 3
**Rating:** 4
**Confidence:** 4

**Summary:**

This paper presents PROMETHEUS, a multi-agent system designed to automate the resolution of GitHub issues across multilingual codebases. The core contribution is the transformation of an entire code repository into a "unified knowledge graph" (KG) that models files, shallow Abstract Syntax Trees (ASTs), and text chunks. This KG, persisted in Neo4j, is used by a multi-agent workflow to perform issue classification, bug reproduction, context retrieval, patch generation, and verification . The system is evaluated on SWE-bench Lite, where it achieves a 35.33% resolution rate, and on the new SWE-bench Multilingual benchmark, where it resolves 25.7% of issues across seven languages. The authors also test the system on eight live issues from two open-source projects, resolving four. The paper claims novelty in its KG-based approach, its demonstrated multilingual capabilities, and its applicability to real-world issues beyond pre-configured benchmarks.

However, these claims are not fully substantiated. The "unified knowledge graph" is a structurally simplistic file index with shallow ASTs (depth 0 and 1), and the paper lacks a critical ablation study to prove this KG is more effective than standard retrieval methods using the same DeepSeek-V3 model. Furthermore, the system's multilingual support relies on the LLM, not the language-agnostic graph structure, and the claim of "real-world" effectiveness is based on an insufficient sample (N=8) where the system failed on half the tasks.

**Strengths:**

1). The design of node/edge types is clear and general across languages

2). The decomposition into Issue Classification, Context Retrieval, Bug Reproduction, Patch Generation, Patch Verification, and Issue Response is understandable and reasonably modular.

3). The paper reports API cost for the 300-instance runs and claims favorable cost versus a GPT-4o baseline. Such transparency is helpful for the community.

**Weaknesses:**

1). The paper's primary claim is that its "unified knowledge graph" is the key component enabling superior performance. However, this claim is left unsubstantiated due to a critical lack of proper ablation studies. The main performance comparison (Figure 2) pits PROMETHEUS (using DeepSeek-V3) against baselines like Agentless (using GPT-40) and AutoCodeRover (using GPT-4). Yet, this is not an apples-to-apples comparison. The reported 35.33% success rate is confounded by the choice of the underlying LLM. It is impossible to determine if the performance gain comes from the proposed KG-based retrieval or simply from the DeepSeek-V3 model's superior capabilities on this task compared to GPT-40. To validate its core contribution, the paper must include a baseline that uses the same DeepSeek-V3 model in an "Agentless" setup (e.g., with a standard BM25 or vector-based retrieval) instead of the KG. Without this direct comparison, the central hypothesis—that the KG is a better context retrieval mechanism—is unproven.

2). The paper's "unified knowledge graph" is presented as a sophisticated model for codebase understanding, but its actual implementation is highly simplistic. The graph consists of only three node types (FileNode, ASTNode, TextNode) and five basic, structural edge types (HAS_FILE, HAS_AST, PARENT_OF, HAS_TEXT, NEXT_CHUNK). Where the AST parsing is "intentionally shallow, focusing only on nodes at depths 0 and 1". This representation lacks the deep, semantic relationships (e.g., function calls, inheritance, dataflow, "implements" relationships) that one would expect from a "knowledge graph" for code. The proposed graph is, in effect, a structured file index combined with shallow ASTs, which is a minor variation on existing code-graph techniques (like the cited RepoGraph ).

3). The paper claims that the KG is what "support multiple programming languages". This is misleading. The graph structure itself is language-agnostic; it simply stores files and Tree-sitter AST nodes (which is inherently language-agnostic). The entire burden of understanding the syntax, semantics, standard libraries, and idioms of seven different programming languages (Java, Rust, C/C++, etc. ) falls on the LLM (DeepSeek-V3). The paper provides no evidence that the graph aids in multilingual resolution in any meaningful, language-specific way. The results in Table 1, which show high variance (e.g., 34.9% for Java vs. 6.8% for Ruby ), are far more likely to be a reflection of the LLM's uneven training data for those languages than a property of the graph architecture.

4). The paper's claim to "resolve real-world GitHub issues in the LangChain and OpenHands repositories"  is based on an evaluation that is too small to be meaningful enough. The evaluation was conducted on a total of only eight issues, of which the system successfully resolved four. Making claims of real-world applicability based on a sample size of N=8 is a significant overstatement. Furthermore, the paper lists the failure modes for the other half of these real-world issues: "API rate limits," "output does not always conform to the expected format," and "context window limitations" . These are not minor issues; they are fundamental failures of the system's robustness and scalability, directly contradicting the claim that it is applicable to "real-world" scenarios.

5). The paper states both “resolves 28.7% (86/300)” and “pass@1 35.33%” for SWE-bench Lite. It is not clear whether “resolve” differs from pass@1, how they are computed, or why both numbers serve as headline results.

**Questions:**

1). What is the precise definition of “resolves 28.7% (86/300)” vs. “pass@1 35.33%”? Which is the main metric? Were both computed with the official SWE-bench evaluator, and under what constraints?

2). Is the total 41/300 or 77/300? Please correct the manuscript and provide a CSV of instance-level results and evaluator logs.

3). The current text both requires matching the expected error and, paradoxically, states that a different error still counts as “correct reproduction.” Which rule is used in code? Provide the exact check.

4). Show results for: no-graph retrieval; file-only retrieval; shallow vs. deeper AST; and graph vs. embedding-only retrieval. This is essential to validate the contribution.

5). Were all compared systems permitted web search and repository-wide exploration? Normalize per-issue tool/compute/token budgets or report a cost-adjusted comparison table.

6). How do you prevent infinite retrieval/refinement loops? Provide controller pseudo-code, timeouts, and their effect on success rate and cost.

7). Please formalize the metric, explain how “plausible patches” are labeled, and justify its interpretability for cross-paper comparison.

---

### Official Review · Reviewer_973F · 2025-11-01

**Soundness:** 3
**Presentation:** 3
**Contribution:** 2
**Rating:** 4
**Confidence:** 3

**Summary:**

The paper introduces PROMETHEUS, a multi-agent system that converts a repository into a unified knowledge graph (files, shallow AST nodes, and text with five edge types, persisted in Neo4j) to drive multilingual issue resolution, reporting 35.33% pass@1 on SWE-bench Lite and 25.7% on SWE-bench Multilingual.

**Strengths:**

- The paper cleanly formulates repository-scale, multilingual issue resolution via a unified KG that spans files, AST fragments, and documentation, and wires this into a practical multi-agent pipeline (classification → reproduction → retrieval → patching → verification → response). This design is clearly motivated and depicted.

- The node/edge schema is explicit (File/AST/Text; HAS_FILE / HAS_AST / PARENT_OF / HAS_TEXT / NEXT_CHUNK), with implementation details and persistence in Neo4j; the shallow AST policy is stated up front, which helps reproducibility and future extension.

- PROMETHEUS is among the first to report SWE-bench Multilingual results across seven languages and shows cost-effective operation (e.g., $0.23 / $0.38 per issue) and some real-world issue resolutions, indicating potential for deployment beyond sandboxed containers.

**Weaknesses:**

- Positioning vs. prior code-graph/RAG work is under-developed. The paper should more sharply differentiate the KG (schema, multilingual alignment, and retrieval) from prior repository-level code graphs (e.g., RepoGraph) and other agentic systems; right now, the narrative asserts Python-only scope for some baselines but lacks controlled, head-to-head comparisons on matched tasks.

- Representation capacity may be too shallow. AST nodes are only kept at depth 0–1; edges encode structure and text adjacency but omit semantic relations (calls, dataflow, symbol resolution). This likely limits cross-file reasoning and could explain retrieval loops/failures on large repos; ablations with deeper ASTs or semantic edges would strengthen claims.

**Questions:**

What truly drives the gains? Could you report edge-type (e.g., remove HAS_TEXT, add call/dataflow) and AST-depth ablations, plus retrieval-loop diagnostics, to tie improvements to specific graph choices and to large-repo behavior?

---

### Meta-Review · Area_Chair_w3qe · 2025-12-31

**Summary:**

The paper proposes an agent scaffold for a software engineering agent with the particular focus on a novel retrieval component that constructs a graph knowledge base connecting files and AST nodes. The system leverages DeepSeek-V1 and achieves comparable performance to other agent scaffolds using GPT-4o.

**Reviewer Concerns:**

1. The model primarily compares to models that perform quite poorly on SWE-bench, as opposed to for example GPT-5, Sonnet-4, Sonnet-4.5, Qwen-Coder. This makes it hard to understand whether the context retrieval component is relevant with state-of-the-art approaches.
2. The claim of “novel” solved issues is concerning since (a) it does not seem to consider more recent approaches and (b) the pass/fail of issues is inherently stochastic and could be due to random sampling (in the absence of pass@k analysis).
3. Representation capacity may be too shallow

**Reviewer Scores:**

The paper received all negative ratings. The authors have not provided the rebuttal.

---

### Decision · Program_Chairs · 2026-01-26

Reject